# Bicoid gradient formation mechanism and dynamics revealed by protein lifetime analysis

Lucia Durrieu[1,2,†,‡], Daniel Kirrmaier[1,3], Tatjana Schneidt[2], Ilia Kats[1] (iD), Sarada Raghavan[1,§], Lars Hufnagel[2,*] (iD), Timothy E Saunders[2,4,5,**] & Michael Knop[1,3,***] (iD)

## Abstract

Embryogenesis relies on instructions provided by spatially organized signaling molecules known as morphogens. Understanding the principles behind morphogen distribution and how cells interpret locally this information remains a major challenge in developmental biology. Here, we introduce morphogen-age measurements as a novel approach to test models of morphogen gradient formation. Using a tandem fluorescent timer as a protein age sensor, we find a gradient of increasing age of Bicoid along the anterior–posterior axis in the early *Drosophila* embryo. Quantitative analysis of the protein age distribution across the embryo reveals that the synthesis–diffusion–degradation model is the most likely model underlying Bicoid gradient formation, and rules out other hypotheses for gradient formation. Moreover, we show that the timer can detect transitions in the dynamics associated with syncytial cellularization. Our results provide new insight into Bicoid gradient formation and demonstrate how morphogen-age information can complement knowledge about movement, abundance, and distribution, which should be widely applicable to other systems.

**Keywords** *Drosophila melanogaster*; embryogenesis; fluorescent timers; morphogen gradient; SPIM

**Subject Categories** Development & Differentiation; Quantitative Biology & Dynamical Systems

**Mol Syst Biol. (2018) 14: e8355**

## Introduction

Acquisition of different cell fates at specific spatial and temporal locations is an essential process driving development. The necessary information is provided locally by morphogens (Wolpert, 1969; Lander, 2011). Understanding morphogen gradient formation requires systematic measurement of the morphogen abundance, mobility, and distribution using temporally resolved methods. However, the technical challenges associated with this undertaking are high, leading to significant discussions on how to best assess the principles and mechanisms (Ribes & Briscoe, 2009; Rogers & Schier, 2011; Muller *et al*, 2013) that have resulted in a plethora of models for the formation of morphogen gradients.

In the early fly embryo, the morphogen protein Bicoid (Bcd) forms a concentration gradient along the anterior–posterior (AP) axis of the embryo, triggering differential cell fate acquisition (Driever & Nüsslein-Volhard, 1988a,b) (Fig 1A). The process is initiated during oogenesis where Bcd mRNA (*bcd*) is localized to the anterior of the forming embryo (Frigerio *et al*, 1986; Berleth *et al*, 1988; Ribes & Briscoe, 2009; Rogers & Schier, 2011; Muller *et al*, 2013). The classic view of Bcd gradient formation is that the protein is synthesized in the anterior pole of the *Drosophila* blastoderm and forms a long range gradient through diffusion, with the gradient shape adapted by protein degradation (SDD model) (Driever & Nüsslein-Volhard, 1988a,b; Gregor *et al*, 2007b). Such a model agrees well with the observed Bcd gradient in embryos undergoing cellularization, where Bcd levels decay exponentially toward the posterior pole (Houchmandzadeh *et al*, 2002; Gregor *et al*, 2007b). However, several other models involving alternative mechanisms for Bcd production and distribution have been proposed, all of which are capable of producing an exponential-like concentration profile, as further outlined below (Coppey *et al*, 2007; Hecht *et al*, 2009; Spirov *et al*, 2009; Dilão & Muraro, 2010; Grimm *et al*, 2010; Kavousanakis *et al*, 2010).

Efforts to distinguish experimentally the different mechanisms of gradient formation have been hindered by uncertainties associated with the measurements of the relevant parameters: local production rates of Bcd; Bcd mobility and transport; and Bcd degradation.

1 Zentrum für Molekulare Biologie der Universität Heidelberg (ZMBH), DKFZ-ZMBH Alliance, University of Heidelberg, Heidelberg, Germany
2 European Molecular Biology Laboratory (EMBL), Heidelberg, Germany
3 Deutsches Krebsforschungszentrum (DKFZ), DKFZ-ZMBH Alliance, Heidelberg, Germany
4 Mechanobiology Institute and Department of Biological Sciences, National University of Singapore, Singapore
5 Institute of Molecular and Cell Biology, A*Star, Biopolis, Singapore
 *Corresponding author. Tel: +49 6221 387 8648; E-mail: hufnagel@embl.de
 **Corresponding author. Tel: +65 66011552; E-mail: dbsste@nus.edu.sg
 ***Corresponding author. Tel: +49 6221 54 4213; E-mail: m.knop@zmbh.uni-heidelberg.de
 †Present address: Instituto Leloir, Buenos Aires, Argentina
 ‡Present address: Departamento de Fisiología, Biología Molecular, y Celular, Facultad de Ciencias Exactas y Naturales, Universidad de Buenos Aires, Buenos Aires, Argentina
 §Present address: p53 Laboratory, A*STAR, Singapore

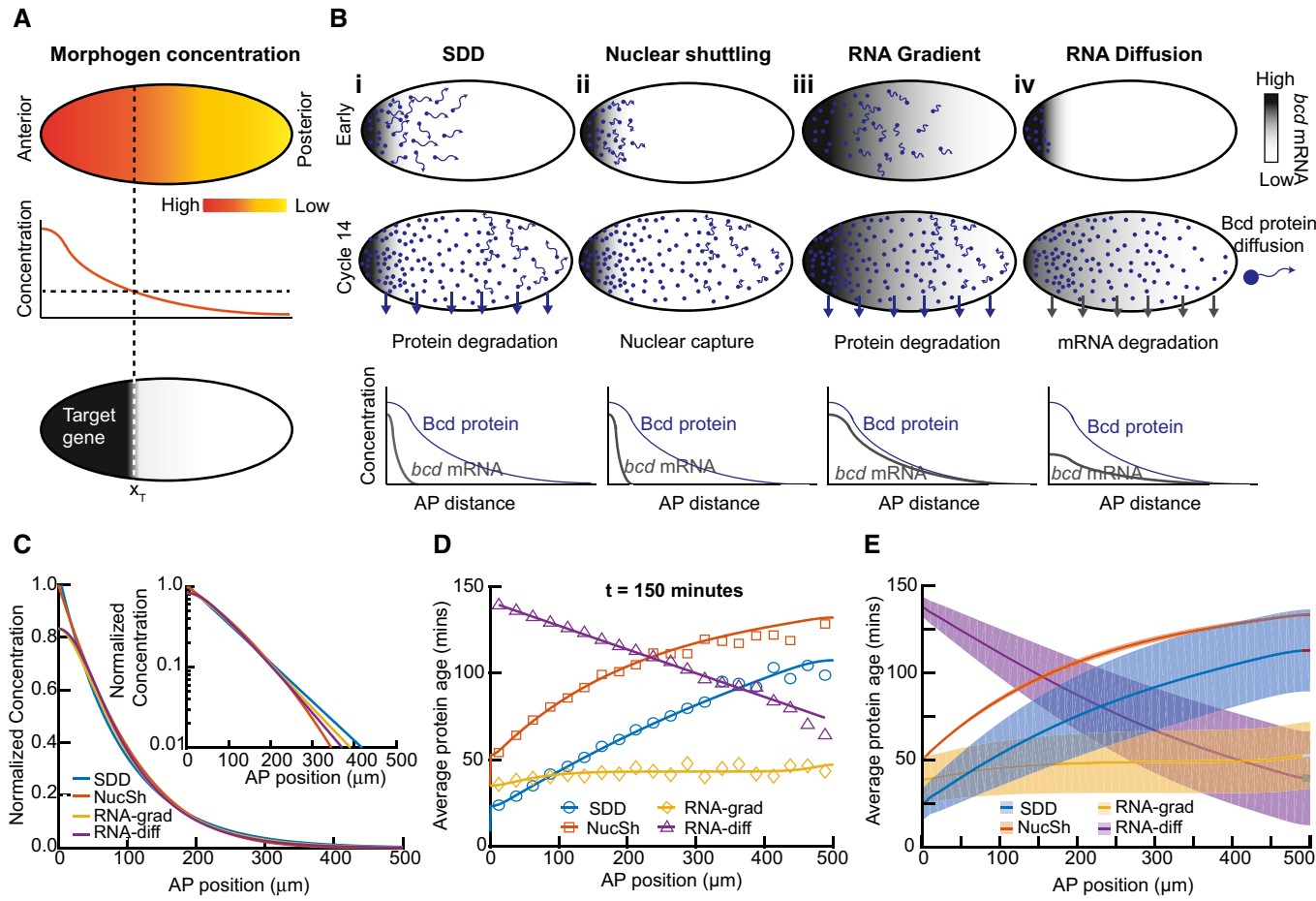

**Figure 1.  Protein age can distinguish alternative models of morphogen gradient formation with similar concentration profiles.**

A   Cartoon of the morphogen hypothesis: a spatially varying concentration of signaling molecule can result in precise readout of positional information.

B   Outline of models considered. Morphogen RNA (grays) and protein (green dots) distribution are shown in early (top) and cycle 14 (middle) rows for each model considered. The magnitude of protein movement is represented by length of green arrows. Degradation of protein/RNA is represented by green/gray arrows. Bottom row shows a schematic of morphogen RNA and protein concentration profiles in cycle 14.

C   Normalized Bcd concentration profiles for the models considered in (B) at time = 2.5 h. Inset: same on log scale. See Appendix and Appendix Fig S1 for extended discussion of models and parameters.

D   Data points correspond to mean output of stochastic Monte Carlo simulation results for the average protein age as function of position, 2.5 h after initiation (see Appendix and Materials and Methods for details). Colored lines correspond to theoretical predictions for protein age in each model (Appendix Section B).

E   As (D) but showing solutions over larger parameter space. Solid line represents mean solution, and dashed lines represent 1 SD. Parameter range described in text. SDD: synthesis, diffusion, degradation model, NucSh: nuclear shuttling model, RNA-grad: RNA gradient model, and RNA-diff: RNA diffusion model.

Experimental estimates of the diffusion constant vary by an order of magnitude (Gregor *et al*, 2007b; Abu-Arish *et al*, 2010), although recent theoretical work has attempted to meld these measurements (Castle *et al*, 2011; Sigaut *et al*, 2014). Estimations of the Bcd degradation rate also significantly differ (Drocco *et al*, 2011; Liu & Ma, 2011; Liu *et al*, 2011). Finally, the extent of the region where Bcd protein is produced is unclear (Spirov *et al*, 2009; Little *et al*, 2011), with a long-ranged gradient of *bcd* mRNA possibly enabling local translation of Bcd away from the anterior pole of the embryo (Spirov *et al*, 2009).

These differences in the diffusion and degradation rates are meaningful beyond the determination of the gradient formation mechanism, as they help to predict whether the system is in (or close to) equilibrium—a contested issue relevant to the mechanism of gradient interpretation (Bergmann *et al*, 2007, 2008; Bialek *et al*,

2008; Saunders & Howard, 2009; Jaeger, 2010; de Lachapelle & Bergmann, 2010a,b). Altogether, these debates regarding nearly every aspect of Bcd gradient formation argue for the need of more incisive tools to investigate this paradigmatic problem.

In this work, we revisit Bcd gradient formation motivated by the observation that tagging of Bcd with different fluorophores results in changes in the apparent gradient shape (Little *et al*, 2011; Wieschaus, 2016). Differences in the fluorophore maturation rates could underlie this change (Little *et al*, 2011; Wieschaus, 2016). We employ the tandem fluorescent protein timer (tFT) reporter (Khmelinskii *et al*, 2012; Donà *et al*, 2014) fused to Bcd (we henceforth refer to this reporter as tFT-Bcd), which provides simultaneously quantitative information about the Bcd protein age and its spatio-temporal concentration distribution. We pair it with multi-view light-sheet fluorescent microscopy (Krzic *et al*, 2012; Tomer *et al*, 2012) to gain

high spatial and temporal resolution images of the tFT-Bcd gradient *in toto*. These data are then used to discriminate between alternative models of Bcd gradient formation, estimate dynamic parameters, investigate the mechanism of Bcd degradation, and to study temporal changes through the early fly embryogenesis.

# Results

### Protein age can distinguish different models of Bcd gradient formation

Broadly, two types of models have been considered for Bcd gradient formation: localized Bcd synthesis in the anterior with subsequent long-ranged transport; and pre-patterned synthesis (by an mRNA gradient) and restricted protein transport, although more complicated scenarios, such as spatially patterned degradation, are imaginable. Interestingly, protein degradation is not a mandatory ingredient in gradient formation, but it plays an important role in determining whether the system can reach steady state. Within this framework, we consider four models: the SDD model (Driever & Nüsslein-Volhard, 1988a; Gregor *et al*, 2007b); the *nuclear shuttling model* (Coppey *et al*, 2007); *bcd mRNA gradient* (Spirov *et al*, 2009; Grimm *et al*, 2010; Dalessi *et al*, 2012); and *bcd mRNA diffusion and degradation* (Dilão & Muraro, 2010; Dalessi *et al*, 2012) (summarized in Fig 1A and B and Appendix Fig S1A). In the SDD and nuclear shuttling models, protein is synthesized locally and then migrates by diffusion toward the posterior pole (Fig 1B, i-ii). The SDD model incorporates protein degradation, whereas the nuclear shuttling model utilizes the rapid increase in nuclei number in the blastoderm to enable the Bcd concentration to remain roughly constant in each nucleus. The RNA gradient model (Fig 1B, iii) is based on a spatially extended RNA gradient resulting in distributed protein synthesis and incorporates protein degradation and very slow protein diffusion. The RNA diffusion model starts with localized RNA and protein synthesis and then proposes spreading of the mRNA (protein synthesis) throughout the embryo (Fig 1B, iv). This model reaches a steady state when the RNA is completely degraded, even though Bcd protein does not decay (and the Bcd protein cannot diffuse) (Dilão & Muraro, 2010). Further model details are provided in the Appendix.

All four models can reproduce the observed Bcd concentration profile at nuclear division cycle (n.c.) 14, as expected from previous reports (Fig 1C, Appendix Fig S1B). Distinguishing these models experimentally then requires precise measurement of dynamic parameters—which has proven challenging so far. Thus, easily measurable information is required where the models make distinct and robust predictions. What information could this be?

It has been shown that tandem fusions of two different fluorescent proteins with different maturation rates can measure the average time that has passed since the production of a pool of proteins (*i.e.*, its age) (Khmelinskii *et al*, 2012). Protein age is dependent on protein turnover and degradation. We reasoned that fluorescent timers could be a valuable tool for discerning the above models experimentally, which prompted us to explore their predictions regarding protein age of Bcd (Appendix Fig S1C). In models that include degradation, the average age of Bcd approaches a steady-state situation where synthesis and degradation are balanced, while in models without protein degradation, the average protein age is constantly increasing (Appendix Fig S1D). We calculated the Bcd protein age as a function of position along the AP axis (Fig 1D, Appendix Fig S1D and E, and Appendix Section B). For the SDD model and the nuclear shuttling models, protein age increases with the distance from the anterior pole, though the average protein age in the SDD model is lower due to degradation. For the RNA gradient model—where the *bcd* RNA gradient is the primary determinant of the Bcd gradient—the average age of Bcd is roughly uniform. In contrast, the RNA diffusion model makes an inverse prediction, with lower protein age toward the posterior pole (Appendix Fig S1E). Therefore, measurement of protein age offers a quantitative readout that can clearly distinguish different models without a need for detailed parameter estimates.

To test the robustness of these predictions, we explored the parameter space underlying each model for the situation 2.5 h after egg laying (AEL), around early n.c. 14. We varied each parameter over a physiologically relevant range: diffusion coefficient (0.1–10 $\mu m^2$/s), protein and RNA lifetimes (10–120 min), and the range of the RNA gradient (20–200 $\mu m$). We only considered parameter sets that resulted in a Bcd gradient with an exponential-like profile with decay length ($\lambda$) between 70 and 100 $\mu m$ 2.5 h AEL (determined by fitting the simulated profiles to $Ae^{-x/\lambda}$ in the range 100–400 $\mu m$ from the anterior pole). This revealed that the principal difference of the models with respect to relative Bcd protein age in the gradient is robust, and should allow faithful discrimination between the models (Fig 1E and Appendix Fig S1F). We note that distinguishing the SDD and nuclear shuttling models is the most challenging. However, combining the protein age data with the protein concentration profiles should enable discrimination between the two models. The shuttling model can only produce an exponential gradient in early n.c. 14 with a limited range of parameters. This constraint is the reason for the narrower range of predictions of the nuclear shuttling model (Fig 1E).

### Establishment of a protein age reporter line

Having established protein age as critical information for the discrimination of the alternative models of Bcd gradient formation, we proceeded to estimate it experimentally. We tagged Bcd with tandem fluorescent timer (tFT) reporters, consisting of a fast-maturing and a slow-maturing fluorescent protein separated by a linker (Fig 2A and B). Such tFT reporters can be used to infer average protein age (Khmelinskii *et al*, 2012; Khmelinskii & Knop, 2014). If a protein of interest is tagged with a fast-maturing green fluorescent protein and a slow-maturing red fluorescent protein, a pool of newly synthesized protein will be mostly green, while older proteins will fluoresce in both green and red. Likewise, in steady state, rapid protein turnover results in fewer proteins being fluorescent in the red channel. Therefore, the average age of a pool of proteins tagged with a tFT reporter can be estimated from the ratio of the fluorescence intensities of the fluorophores.

In order to estimate the protein average age from tFT reporters, the timescale for the maturation of the slower maturing fluorophore should be no slower than the dynamics of the system studied (Khmelinskii *et al*, 2012; Khmelinskii & Knop, 2014), which, in our case, is in the order of an hour. Due to uncertainty about both Bcd degradation kinetics and the fluorophores maturation rates in fly

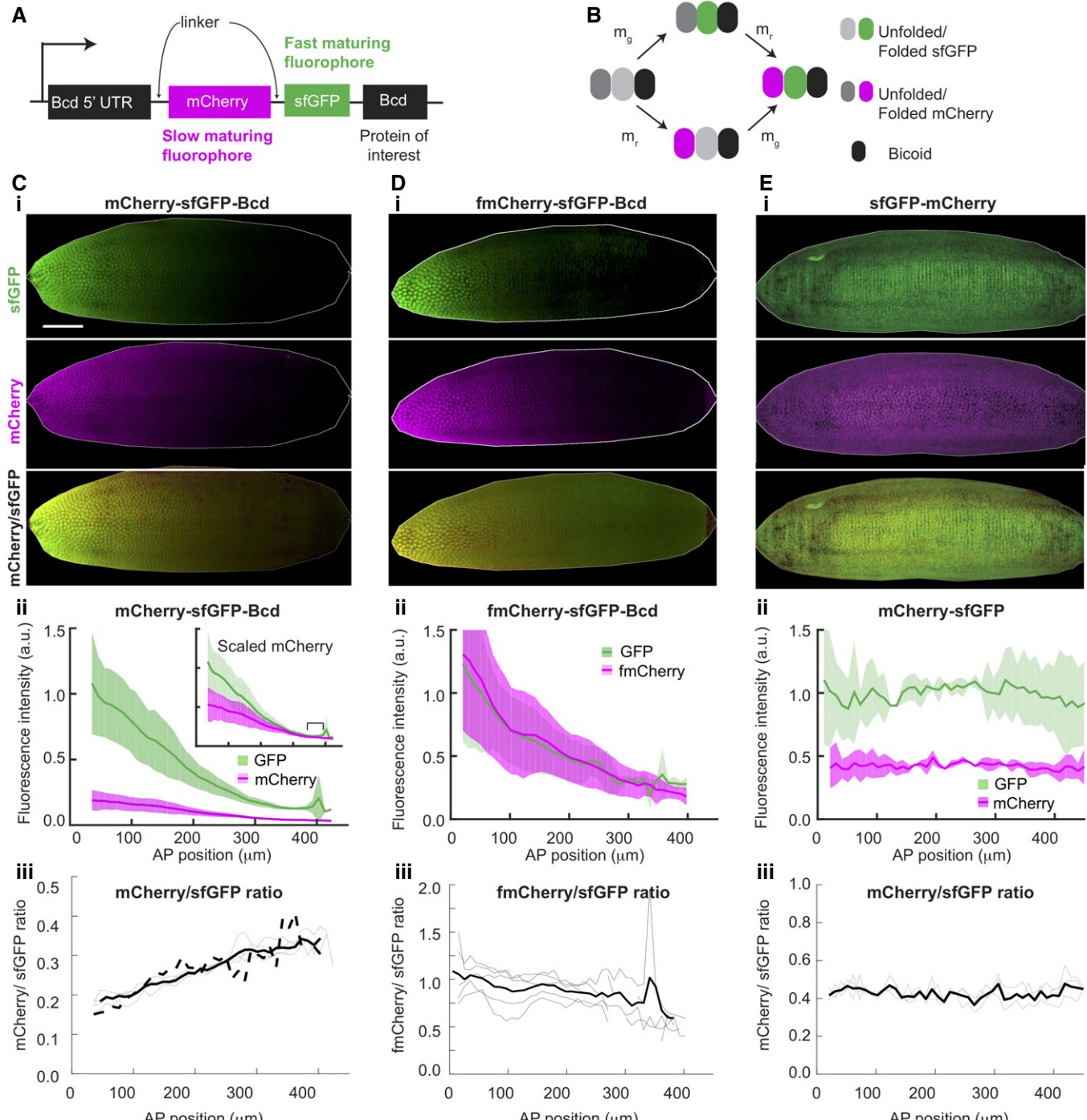

**Figure 2. tFT-Bcd reporter reflects average protein age.**

A    Schematic of tFT-Bcd reporter with mCherry and sfGFP fluorophores.

B    Cartoon of the fluorescent protein maturation states in the tFT-Bcd reporter. $m_g$ and $m_r$ represent the sfGFP and mCherry maturation rates, respectively (note, $m_r$ is an effective rate as mCherry has a two-step maturation process).

C–E    Examples of embryos expressing the tFT-Bcd reporter with (C) mCherry-sfGFP-Bcd, (D) fmCherry-sfGFP-Bcd, and (E) mCherry-sfGFP. For all, (i) images of "shells" of embryos in early n.c. 14. 3D images of the embryos were generated, and then, the interior of the embryo was erased, leaving only the embryo cortex to improve clarity (see Materials and Methods for more details). (ii) Mean AP intensity profile of each color for the embryo in (i). Shade region represents ± 1 SD. Inset shows same profiles after multiplication of red intensity by constant factor. Data binned into 10-μm bins (*n* = 4 embryos). (iii) Ratio of green over red signal, reflecting protein age. The thin lines represent individual embryos, while the thick solid line is the mean. The solid dashed line depicts the mean green/red ratio for a line with the tFT-Bcd reporter but lacking endogenous Bcd (Bcd[E1] mutant, *n* = 4. The scale is the same for all embryos. The scale bar is 50 μm long).

Source data are available online for this figure.

embryos, knowing *a priori* which fluorophore pair to use is difficult. We constructed six tFT-Bcd lines, each containing sfGFP and a different red FP (namely mCherry, tdTomato, fmCherry, td-fmCherry, mKate2, and tagRFP) and analyzed the red and green signal intensity along the AP axis (Appendix Fig S2). fmCherry stands for "fast-maturing mCherry" and was developed using directed protein evolution seeking for fast-maturing variants of mCherry (Materials and Methods). In all constructs, the tFT reporter was fused to the N-terminus of Bcd, keeping the promoter and the 5′ and 3′ UTR of the *bcd* gene (Fig 2A). We validated the functionality of the tFT-Bcd fusions by rescuing the $bcd^{E1}/bcd^{E1}$ null mutant (Driever & Nüsslein-Volhard, 1988a) (Appendix Fig S3). Based on this screening, we chose the lines containing the sfGFP-mCherry (Fig 2C) and sfGFP-fmCherry (Fig 2D) tFT reporters for further investigation. These timers are informative since they display a gradient in both colors, but with different profiles.

Quantitative utilization of the fluorescent timer requires additional controls. It is critical that the age of the tFT-Bcd proteins reflects the stability of Bcd rather than the lifetime of the tFT tag, or a combination of both. To verify that this is the case, we established a control line that expresses only the tFT reporter (without the Bcd protein), but still using the regulatory sequences of the *bcd* gene (Fig 2E). This construct shows similar (flat) profiles for both fluorophores, suggesting: (i) degradation is determined by the Bcd protein, not the linked fluorophores; and (ii) that the measured differences are determined by Bcd dynamics and not the degradation of the fluorescent reporters themselves.

A further potential issue with tFT reporter measurements is incomplete degradation of the fluorescent timer (Khmelinskii *et al*, 2012, 2016). If some fraction of free GFP protein survives the degradation of the tFT-Bcd protein, then the ratio measurements would be affected. We assessed this possibility experimentally by making Western blots against GFP for embryos of different ages (Appendix Fig S4A and B) and theoretically by simulating the impact of such artifact on the observed Bcd gradients (Appendix Fig S4C and D). Both approaches suggest that this issue is likely unimportant here. Since we introduced our constructs into the fly genome as an additional copy of Bcd, another concern is whether the higher protein levels alter the Bcd dynamics. We repeated our tFT-Bcd measurements in embryos with a Bcd null background ($bcd^{E1}$) and found no difference in the tFT reporter readout (Fig 2C iii, dashed line).

To summarize, we have demonstrated that: (i) the Bcd-tFT reporter is functional; (ii) the spatial distribution of the fluorescent signals is dependent on Bcd, and not on the properties of the fluorescent molecules *per se*; and (iii) by tuning the maturation rates of the different fluorophores, we can alter the shape of the fluorescent profiles, supporting the conclusion that the tFT reporter is sensitive to specific time scales. Therefore, we are confident that the tFT reporter does indeed reflect Bcd protein dynamics.

### Imaging and quantification of the tFT-Bcd signal in live embryos

Reliable quantification of fluorescence signals in the *Drosophila* embryo is challenging, mostly due to the relatively large size of the embryo and autofluorescence from the yolk. We utilized confocal multi-view light-sheet microscopy (MuVi-SPIM) (Baumgart & Kubitscheck, 2012; Krzic *et al*, 2012; de Medeiros *et al*, 2015),

which enables highly sensitive *in toto* fluorescence detection in larger specimens, while bleaching and phototoxicity are strongly reduced compared to confocal detection methods (Stelzer, 2015). MuVi-SPIM imaging produces three-dimensional whole-embryo images with sufficient resolution to observe sub-nuclear details. Since working with the three-dimensional datasets for quantification of the tFT-Bcd signal is challenging, we mapped the three-dimensional data to a two-dimensional projection (Krzic *et al*, 2012) that represents the embryo's cortex, the region of the embryonic periphery where the nuclei—and Bcd—reside (Appendix Fig S5A and B; Materials and Methods). This step significantly reduced the image size and facilitated handling without loss of relevant information (for validation of methodology and microscope sensitivity, see Appendix Fig S5C–F, and the Appendix). MuVi-SPIM imaging enabled us to detect Bcd-eGFP fluorescence signal as early as n.c. 8 (Appendix Fig S5G), and in n.c. 14, we observed intensity variability similar to previously reported fluctuations in Bcd (Gregor *et al*, 2007a) (Appendix Fig S5I) and above-background Bcd signal in all nuclei, even at the posterior pole (Mir *et al*, 2017) (Appendix Fig S5H). This confirms the high sensitivity of detection of the tFT-Bcd signal using the MuVi-SPIM.

### Protein age as an independent test of models of Bcd gradient formation

Quantification of the tFT-Bcd fluorescence in early n.c. 14 embryos revealed that the mCherry/sfGFP intensity ratio increased along the AP axis (Fig 2C iii). This implies that, on average, the tFT-Bcd is older toward the posterior. In contrast, the fmCherry-sfGFP-Bcd line showed a reversed ratio behavior (Fig 2D iii), consistent with fmCherry maturing faster than sfGFP. In conclusion, our quantitative analysis demonstrates an increase in the relative tFT-Bcd age toward the posterior pole. We validated that this result does not depend on the specific scaling of the measured intensities (Appendix Fig S5M and N). Already analyzing these data qualitatively, we can conclude that the tFT-Bcd age profile is not consistent with the predictions from the RNA gradient and RNA diffusion models (Figs 2C iii and 1D and E).

To confirm this preliminary and qualitative interpretation of the data in a quantitative manner and to potentially also discriminate between the SDD and the nuclear shuttling models, more precise estimates of the fluorophore maturation rates are required. Given that fluorophore maturation is sensitive to the experimental conditions, we determined them in the early fly early embryo and in the context of the tFT reporter (Appendix Section D and Appendix Fig S6). We estimated a maturation time of $T = 27 \pm 2$ min for sfGFP. For mCherry, maturation involves two subsequent reactions for which we estimated individual times of $40 \pm 9$ min and $9 \pm 4$ min. fmCherry showed remarkably fast kinetics, with the two steps giving maturation times of each $6 \pm 2$ min (Appendix Fig S6, Appendix). These measured maturation rates are consistent with the observed behavior of the tFT-Bcd reporters in Fig 2C and D.

We fitted the models to the experimentally measured tFT-Bcd gradients from early n.c. 14 (see Materials and Methods for fitting details). The parameters for each model were fitted simultaneously along the AP axis to the Bcd sfGFP profile and the ratio of the two fluorescent signals (Fig 3A, Appendix Fig S7A–H, Materials and Methods). Consistent with the preliminary assessment, we found

that the RNA diffusion model is inconsistent with the data as it predicts a slope for the timer ratio profile opposite to the experimentally determined. For the RNA gradient model (with *bcd* decay length of 100 μm) and the shuttling model, the optimal solutions were able to fit the measured sfGFP profile but not the observed mCherry/sfGFP ratio. Although they both predict aging protein toward the posterior, it is clear that neither model is able to produce a protein age profile similar to observations. As noted above, since

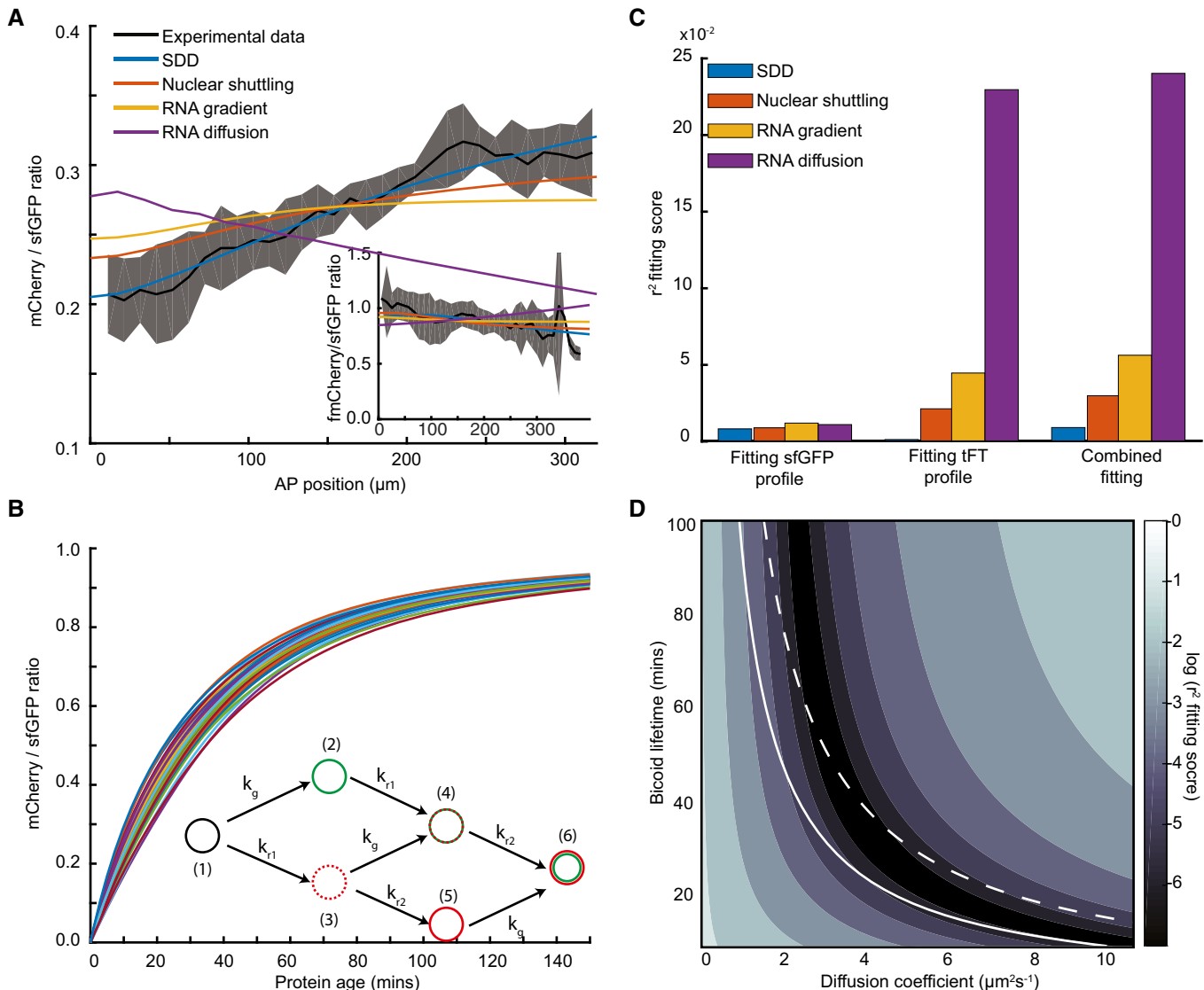

**Figure 3.  The SDD model is most consistent with the tFT-Bcd reporter.**

A   Model fitting to the experimental tFT-Bcd ratio in early n.c. 14 as function of AP position for the different models outlined in Fig 1B. Black line represents mean measured ratio of tFT-Bcd reporter with mCherry-sfGFP-Bcd in y/w flies. The shadowed area represents ± 1 SD. Inset is same as main panel, except for the tFT-sfGFP-fmCherry-Bcd reporter.

B   Increasing protein age results in increasing tandem reporter ratio. The fluorophore states of the tFT reporter are shown in the inset: (i) immature; (ii) only sfGFP matured; (iii) only intermediate mCherry matured; (iv) sfGFP and intermediate mCherry matured; (v) only mCherry matured; and (vi) both mCherry and sfGFP matured. $k_g$, $k_{r1}$, $k_{r2}$ denote the maturation rates of sfGFP, intermediate mCherry, and fully matured mCherry, respectively. The different curves correspond to sets of values for $k_g$, $k_{r1}$, $k_{r2}$ selected from Gaussian distributions with half-time means 27, 40, and 9 min and standard deviations 2.7, 4, and 0.9 min, respectively.

C   Least-squares fitting of each model in n.c. 14 to: (left) the sfGFP only; (middle) the tFT-Bcd reporter ratio only; and (right) both the sfGFP profile and the tFT-Bcd reporter ratio simultaneously. Each fit is performed with the fluorophore maturation rates drawn from Gaussian distributions as in (A). See Materials and Methods for details of fitting.

D   Quality of SDD model fitting to tFT-Bcd ratio (shown in A) for different Bcd diffusion coefficients and lifetimes. For each pair of these parameters, the fluorescence correction ε is left as a free parameter for fitting. The colormap is the log of the fitting function (Materials and Methods). The white lines correspond to parameter combinations that produce a gradient with a length constant λ = 75 μm (solid) and λ = 95 μm (dashed).

Source data are available online for this figure.

the nuclear shuttling model never reaches a steady state, it has a narrow range of parameters that can fit the observed data. The SDD model permits the best general fit to the tFT-Bcd concentration and ratio profiles, being able to capture the experimental data fully (Fig 3A). We then took the fitted parameters for each model and tested each model with the data from the sfGFP-fmCherry tFT reporter. The SDD model again had the best model fit, though the nuclear shuttling and RNA gradient models were also qualitatively able to fit the data (Fig 3A inset). In conclusion, fitting of the tFT reporter profiles supports the SDD model as the most likely mechanism underlying Bcd gradient formation.

Of course, the above conclusions are dependent on the fluorophore behavior. First, in Fig 1D and E we showed how the protein age varies as a function of position in each model. In Fig 3B, we show how the protein age is related to the predicted ratio of the sfGFP-mCherry tFT reporter (tFT reporter states shown in inset and see Appendix for details of the calculations), where we account for uncertainty in the fluorophore folding rates. The variability in the predicted tFT reporter ratio for a specific protein lifetime can be large, which increases uncertainty in our estimates of the dynamic parameters. Second, normalization of the tFT reporter measurements is required in order to relate the measured fluorescence intensities in both channels to the respective fraction of matured fluorescent proteins. This is incorporated in our fittings in Fig 3A by a scaling factor, $\varepsilon$, which is a free parameter in the fitting. Again, this broadens the confidence intervals for the dynamic parameters, but introducing $\varepsilon$ does not fundamentally change the general profile shapes of the different models. Finally, there are potential FRET effects between the two fluorescent molecules, which previously have been estimated to be around 15–20% (Khmelinskii *et al*, 2012; Barry *et al*, 2016). In Fig 3, we do not include these effects in the fits shown, but in Appendix Fig S7I and J, we demonstrate that including such FRET effects do not qualitatively change our conclusions (Materials and Methods and Appendix).

Returning to our model fits in Fig 3A, we see that when comparing the fit quality of all the models to the sfGFP-mCherry tFT reporter, the SDD model is clearly optimal (Fig 3C). Importantly, although uncertainties in $\varepsilon$ and the fluorophore folding rates affect our predictions of the dynamic parameters, they do not alter the fundamental shape of the different model predictions. Our results are broadly consistent with the predominant view that the SDD model is the most likely mechanism of Bcd morphogen gradient formation.

## Simultaneous estimation of Bcd degradation rate and diffusion coefficient

We next explored more thoroughly the parameter space of the SDD model. By fitting both the tFT-Bcd ratio and concentration profiles in early n.c.14 to the SDD model, we estimated the effective Bcd diffusion coefficient and the lifetime independently from the same measurements. Taking the best model fits ($r^2 < 10^{-3}$), we found D = 3.6 ± 0.2 μm²/s and Bcd half-life = 26 ± 3 min, which are consistent with previous reported measurements (Fig 3D). The corresponding decay length of the gradient of 89 ± 7 μm is in the range of previously measured values (Liu *et al*, 2013). However, we note that due to the unknown scaling, $\varepsilon$, between the sfGFP and mCherry fluorescence intensity, FRET effects (Appendix Fig S7I and J), and the error in the fluorophore maturation rates, the range of

possible parameter values is likely larger. Considering parameter ranges that result in a fitting score within 10% of the lowest fit score ($r^2 < 1.1 \times 10^{-3}$), we find a range for the parameters: 2.9 μm²/s < $D_{eff}$ < 4.9 μm²/s; 17 min < Bcd half-life < 37 min, with the constraint that the decay length is 85 μm < $\lambda$ < 97 μm. We note that our estimate of the diffusion coefficient represents an effective diffusion that encapsulates protein diffusion, nuclear shuttling, and binding. Consistent with this, our predicted range for the diffusion coefficient is generally smaller than the one measured by FCS, D~7 μm²/s (Abu-Arish *et al*, 2010) (which measure the freely diffusing fraction on a sub-micrometer length scale).

Including an extended *bcd* RNA source with decay length < 40 μm was also consistent with our data. However, using D ~0.3 μm²/s combined with a long-ranged gradient of *bcd* mRNA—as previously hypothesized (Spirov *et al*, 2009)—cannot reproduce the tFT-Bcd ratio (Appendix Fig S7K and L). Therefore, our data place a limit on the spatial extent of the *bcd* mRNA gradient.

Overall, the tFT-Bcd ratio allows clear discrimination between competing models of morphogen gradient formation. Further, we can simultaneously estimate both diffusion and degradation parameters (although with relatively large uncertainty), and therefore, our analysis constitutes an independent approach, distinct from previous estimations of these parameters.

## Testing of Bcd degradation mechanisms

Our analysis suggests that Bcd degradation is essential for shaping its concentration gradient. However, it is currently not clearly understood how Bcd is degraded. When assessed in embryonic extracts, Bcd degradation was reported to be proteasome dependent, and regulated by the F-box protein Fates shifted (Fsd)—a specificity factor of the E3 ubiquitin ligase SCF (Liu & Ma, 2011).

To further test the proteasome dependency *in vivo,* we injected tFT-Bcd carrying n.c. 14 embryos with the proteasome inhibitor MG132 and imaged them 1 h later (Fig 4A–C). In these embryos, the nuclear division cycle was arrested, indicating the efficacy of the proteasome inhibition. The division cycle arrest was observed across the whole embryo, indicating that the inhibitor was distributed approximately homogeneously, mediated by the small molecular size of the inhibitor. We observed increased amounts of tFT-Bcd and a shift in the timer ratio, indicating the presence of older proteins. We then fitted the proteasome inhibition data to quantify the effect on the Bcd lifetime. We were able to fit both DMSO and MG132 injection data with the same parameters using the SDD model, except changing the Bcd lifetime from 26 (DMSO) to 42 (MG132) min. These observations suggest that Bcd is at least partially subject to proteasomal degradation (Fig 4A–C).

To rule out possible indirect effects of the MG132 inhibitor on the reporter, we injected in parallel embryos expressing the construct with the tFT but without Bcd. This construct is expected to be very stable as degradation signals are absent. Accordingly, inhibition of the proteasomes in these embryos did not increase the total fluorescence levels or their ratio (Appendix Fig S8).

We next analyzed the impact of a mutation in Fates shifted (Fsd) on the tFT-Bcd gradient. For this, we crossed the tFT-Bcd line with a *fsd*-deficient line (Appendix Fig S9). We observed no difference between the Fsd mutant and wt embryos in the total amounts or the profile of the tFT-Bcd gradient (Fig 4D and E), and also, the

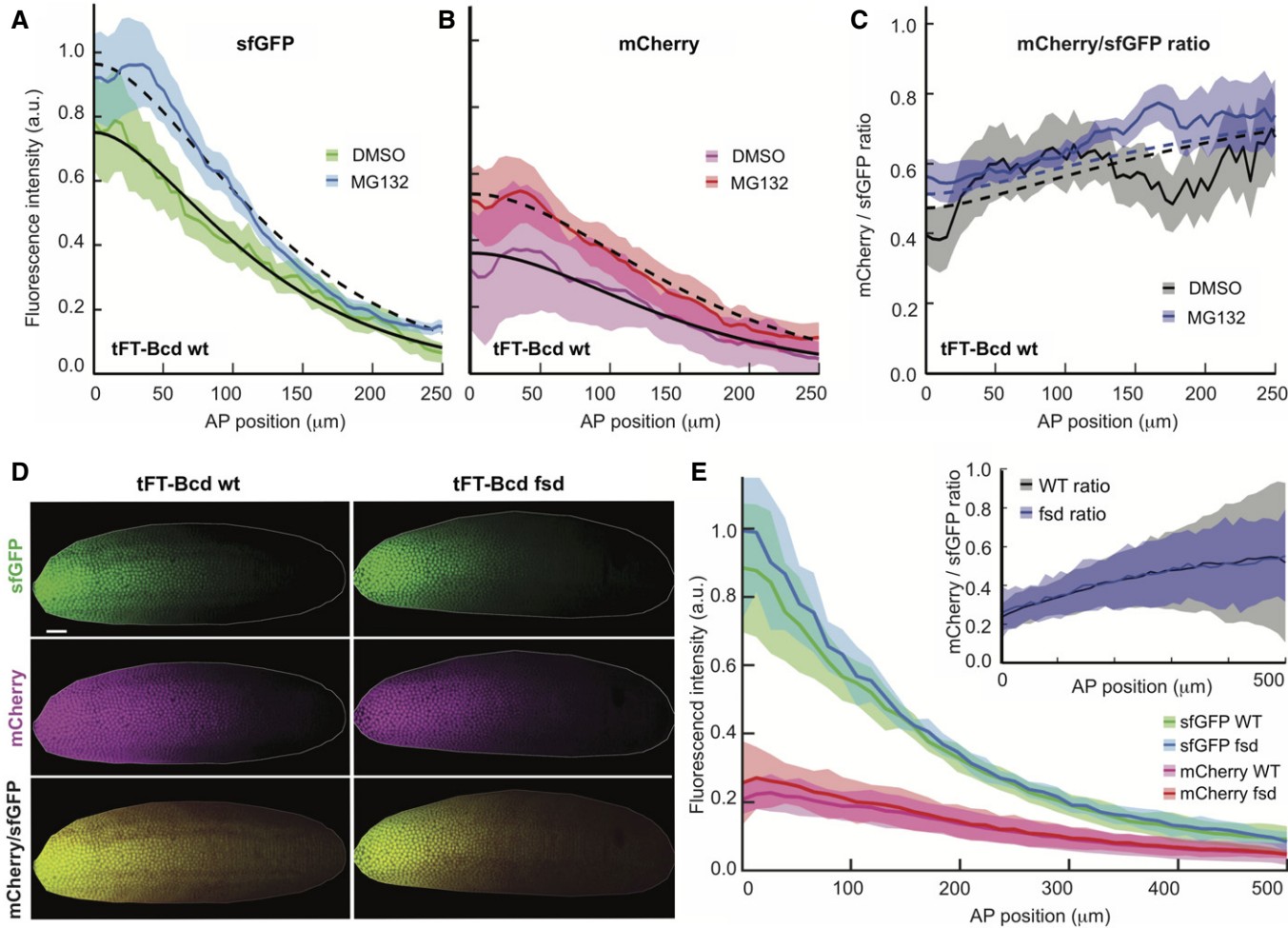

**Figure 4.  Bcd is degraded by the proteasomes *in vivo*.**

A–C    Injection of the proteasome inhibitor MG132 in embryos increases both the amount and the average age of Bcd (gray, open squares, N = 10). Control embryos were injected with DMSO (black, filled circles, N = 9). All embryos are y/w and were injected while in cycle 14 and imaged 60 min later. The shaded regions correspond to the mean ± SD.

D    The age of Bcd and its gradient are unaltered in *fsd* mutant. Comparison of a representative y/w embryo (left) and loss-of-function mutation fsd[KG02393] (right) are shown. Both embryos have the tFT-Bcd construct in the same locus, are in early n.c. 14, were imaged side-by-side under identical conditions, and are displayed with the same settings. Embryo "shells" as in Fig 2 are displayed. The scale bar corresponds to 50 μm.

E    Quantification of embryos as in (D). Fluorescence intensity from sfGFP (green) and mCherry (magenta) for wt (N = 8, filled circles) and fsd (N = 6, open squares) at early cycle 14. Inset: tFT-Bcd ratio for wt (black) and FSD (gray) embryos. The shaded regions depict the mean ± SD.

Source data are available online for this figure.

observed tFT-Bcd ratio was similar in *fsd* and wt embryos in n.c. 14 (Fig 4E inset). This argues against a major role of Fsd in regulating Bcd turnover. Our results do not preclude a role for Fsd in Bcd regulation, but they demonstrate that it is not necessary for Bcd degradation in n.c. 14 and it does not impact Bcd gradient formation during earlier nuclear division cycles.

### Dissecting production and degradation rates using fluorescent timer measurements

So far we have focused on quantifying the tFT-Bcd ratio in early n.c. 14, when the gradient is presumed near equilibrium (Gregor *et al*, 2007b). In steady state, the tFT-Bcd ratio is independent of production and thus gives no information about the Bcd synthesis rate.

However, out of steady state the tFT-Bcd ratio is a function of the production and the degradation rate (Barry *et al*, 2016), see theory section in the Appendix, Section B.5. We next explored the temporal behavior of the tFT-Bcd fluorescence gradient using time-lapse imaging. We acquired 3D movies of developing embryos from the early syncytial blastoderm to gastrulation (Fig 5A, Movie EV1). Light-sheet microscopy minimizes the illumination, and thus the photodamage, of the samples. However, the early embryo is sensitive to phototoxicity thus preventing time-lapse imaging with the same exposures compared to single time point snapshot measurements. Therefore, we needed to reduce the light exposure. As a consequence, the sensitivity of detection was not sufficient for reliable quantification in the posterior half of the embryo and we restricted the analysis to the anterior half of the gradient.

Analysis of the sfGFP intensity profile in these embryos matched previous measurements in fluorescent protein tagged Bcd lines: a clear Bcd gradient along the AP axis by n.c. 10 that increases in intensity until n.c. 13 and then starts to decay during n.c. 13 and n.c. 14 (Fig 5A left, B top) (Bergmann *et al*, 2007; Gregor *et al*, 2007b; Little *et al*, 2011). Looking at mCherry signal alone gave comparable results, although the intensity peak occurs approximately 30 min later, consistent with the longer maturation rate

(Fig 5A center, B middle, C middle). The mCherry/sfGFP ratio suggests that the average Bcd protein age is changing throughout early development (Fig 5A right, B and C bottom), indicating that Bicoid production and degradation are not in steady state.

We next performed *in silico* experiments on the SDD model to explore how changes in production and degradation rates are predicted to alter the tFT reporter readout. To illustrate this analysis, we consider two alternative clearance mechanisms: production

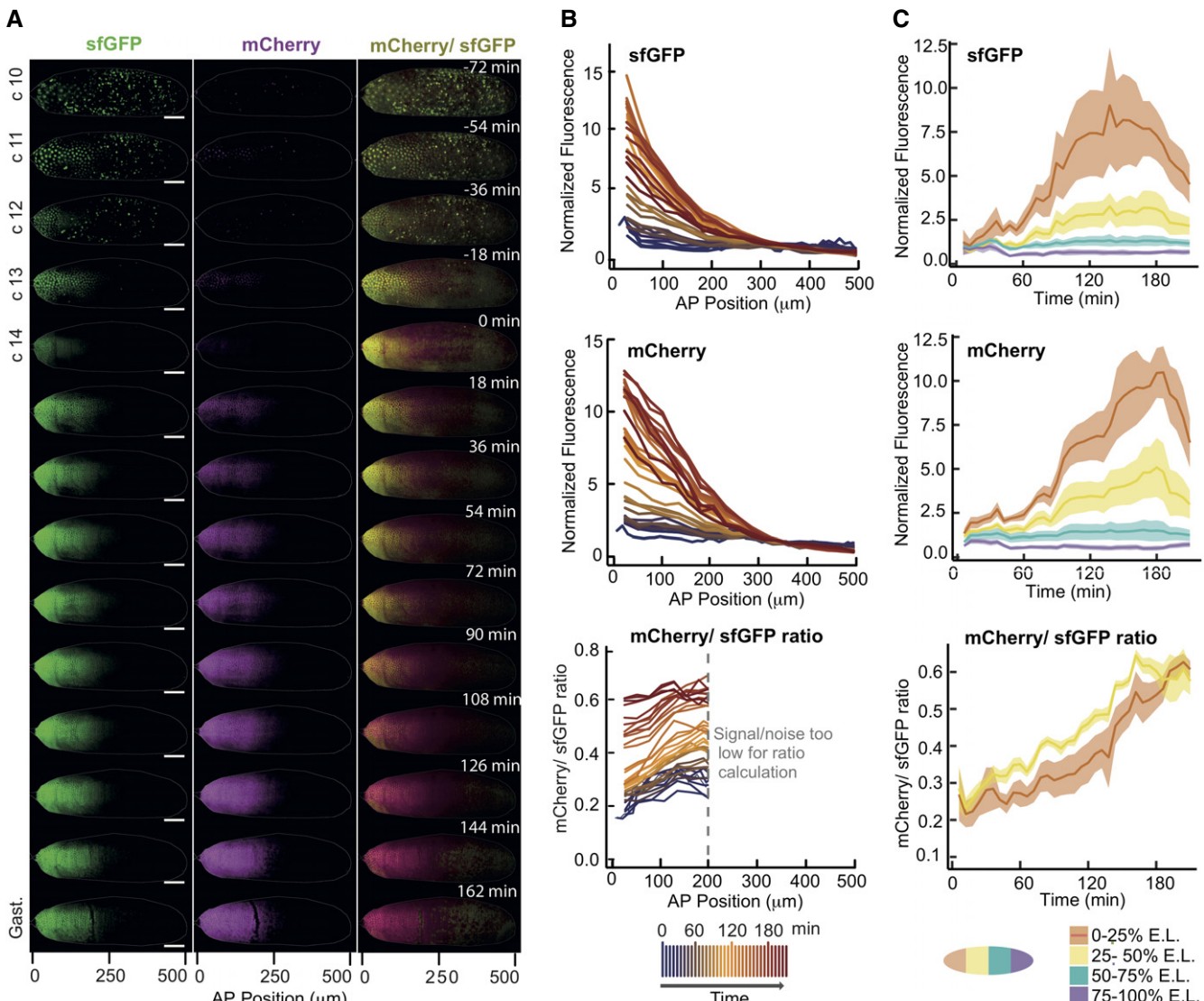

**Figure 5. The average age of Bcd proteins increases continuously during the cleavage divisions and the gastrulation.**

Time-course of a representative y/w embryo with the tFT-Bcd reporter from the division cycle 10 to the early gastrulation. A movie of the same embryo with 6-min time resolution is available as Movie EV1.

A   SPIM images of the embryo early development. Embryo "shells" as in Fig 2 are displayed. Left: Green fluorescence, center: Red fluorescence, right: Intensity-weighted ratiometric image (see Materials and Methods) of the mCherry/sfGFP ratio, reporting average protein age. The scale bar corresponds to 50 μm.

B   Quantification of the same embryo as is (A). Each line represents the fluorescence profile along the anterior–posterior axis of the embryo for a time frame (separated by 6 min). In the lower panel, where the mCherry/sfGFP ratio is shown, the posterior half of the embryo is not shown because the signal drops to the background levels (due to softer imaging conditions for the long term time-course).

C   The same data are shown as a function of time for different embryo segments. The shaded regions show the mean value ± SD.

Source data are available online for this figure.

cessation (with constant degradation) or degradation increase (with constant production). The single-color concentration profiles during the clearance for these two scenarios are largely indistinguishable (Fig 6A insets), but the mCherry/sfGFP ratio increases with time in the first scenario, while it decreases in the second (Fig 6A). Thus, the temporal changes in the tFT ratio can potentially distinguish the alternative scenarios.

When examining Bcd clearance in the experimental time-courses, the observation that Bcd protein age increases constantly through development (Fig 5A right panel, B and C bottom) already discards the scenario with increased degradation and constant production (Fig 6A right panel), since this is predicted to lower the mCherry/sfGFP ratio. This is consistent with the process of cellularization during n.c. 14 acting to isolate local Bcd pools inside cells. Effectively, cells receive fewer new Bcd molecules, at least in the region where there is no *bcd* mRNA.

We extended this analysis to also include concurrent changes in the production and degradation rates. In the simulations, we allowed the system to reach steady state and then changed the production and the degradation rates in a step-wise manner (Fig 6B). In order to visualize the mCherry/sfGFP ratio (age information) and the total intensity (quantity information) simultaneously, we crafted a phase diagram comparing temporal changes in protein age with the total intensity (Fig 6B). The spatial distribution of proteins is not considered for this analysis, only the average values. Due to the imaging limitations outlined above, we considered only the mean Bcd intensity in the anterior region because it has the best signal/noise ratio in the experimental data (other regions give qualitatively similar results, Appendix Fig S10). Different combinations of changes in production and degradation rates resulted in qualitatively distinct trajectories in the phase diagram, as each scenario falls into a different quadrant. This suggests the use of the age-levels analysis can distinguish between different scenarios of how synthesis and degradation change during different phases of development.

### Synthesis–degradation dynamics during early development

In total, we acquired movies from four different embryos which all exhibited similar trajectories (Fig 6C), which we then aligned to each other (Fig 6D). Comparing these experimental trajectories with the theoretical ones from Fig 6B, we propose two main phases: one where the gradient is established (Fig 6D, Phase I); and one where the protein is cleared (Fig 6D, Phase II). In-between, there is a short period (gray circles in Fig 6C) where the system appears to be in a quasi steady state. Afterward, Bcd levels, as measured using the sfGFP signal, appear to stay constant for a short period, while the mCherry/sfGFP ratio starts to increase (Appendix Fig S12). This appears to relate to the phase where Bcd synthesis comes to a halt, and degradation of the protein starts to dominate the Bcd dynamical behavior, resulting in an increase in the tFT reporter ratio followed by a decrease in total fluorescence signal, due to degradation.

## Discussion

How does the Bcd gradient form? Previously, models of Bcd gradient formation were difficult to distinguish experimentally as they all predicted similar concentration profiles in early n.c. 14 and

measurements of dynamic parameters were not sufficiently clear. We have shown here that by measuring the average protein age we can more clearly distinguish these models. Our timer analysis revealed the following dynamic and static constraints underlying Bcd gradient formation: (i) anterior localized Bcd synthesis, with any *bcd* mRNA gradient restricted to < 50 μm; (ii) average Bcd protein diffusion around 2.9–4.9 μm²/s; (iii) spatially homogeneous Bcd degradation with a half-life of about 17–37 min during early n.c. 14; (iv) proteasome (but not Fsd) mediated Bcd degradation; and (v) clearance due to a combination of partially reduced degradation and a dramatically reduced Bcd synthesis during n.c. 14. Our results are most compatible with the SDD model, though the synthesis (and possibly the degradation) parameters are time dependent, as previously suggested (Little *et al*, 2011).

Additionally, we are able to measure the entire extent of the Bcd gradient using MuVi-SPIM. We observe Bcd protein in the posterior pole, a region where mRNA is yet to be observed. This further supports our result that an mRNA gradient plays only a limited role in Bcd protein gradient formation.

Changes in protein levels can be caused by regulation of their synthesis or their degradation. We presented here a new analysis, the tFT reporter in combination with the age/levels diagram, that allows distinguishing them in live, un-perturbed organisms. This approach allowed us to determine that Bcd production is reduced during cycle 14, probably due to degradation of its mRNA or cellularization.

Knowing that Bcd production and degradation do not reach a steady state, and that at least the production rate changes during development, we wondered whether the SDD model could still explain all the dynamics observed. We performed a simulation keeping the SDD model structure, but introducing changes in the production and the degradation rates at a time corresponding to the beginning of n.c. 14 (Appendix Fig S11). The simulation results were capable of reflecting the experimental trajectories closely, with the best fit achieved when stopping production and partially decreasing degradation (Appendix Fig S11). We tested the effect of impeding diffusion during the clearance phase (i.e., due to cellularization), but no appreciable effect on the age/levels diagram was observed. Thus, the SDD model structure, combined with time-dependent parameters, can explain the formation of the gradient during early development.

Motivated by our observation that the kinetic parameters may be time dependent, we revisited the proposed models for Bcd gradient formation incorporating time dependence. One potential mechanism for time dependence is through molecular crowding due to the increased density of nuclei during blastoderm development, which would alter the effective diffusion coefficient. Incorporating a time-dependent diffusion constant, $D = D_0 - D_1(t/t_0)^{1/2}$ ($t_0 = 75$ min) we refitted the nuclear shuttling model to the measured tFT-Bcd profiles in n.c. 14. Though the fit is improved, the nuclear shuttling model is still less good at fitting the data than the SDD model, with both models now having three fitting parameters (Appendix Fig S12). Therefore, we cannot discount such time-varying diffusion, but this requires very precise control of diffusivity.

The underlying mechanisms driving the time dependence in the kinetic parameters could be due to a number of (potentially inter-related) factors. Firstly, the maternal-to-zygotic transition occurs during n.c. 12–14. Therefore, the transcriptional program, including potential degrading factors for Bcd, may be differentially expressed

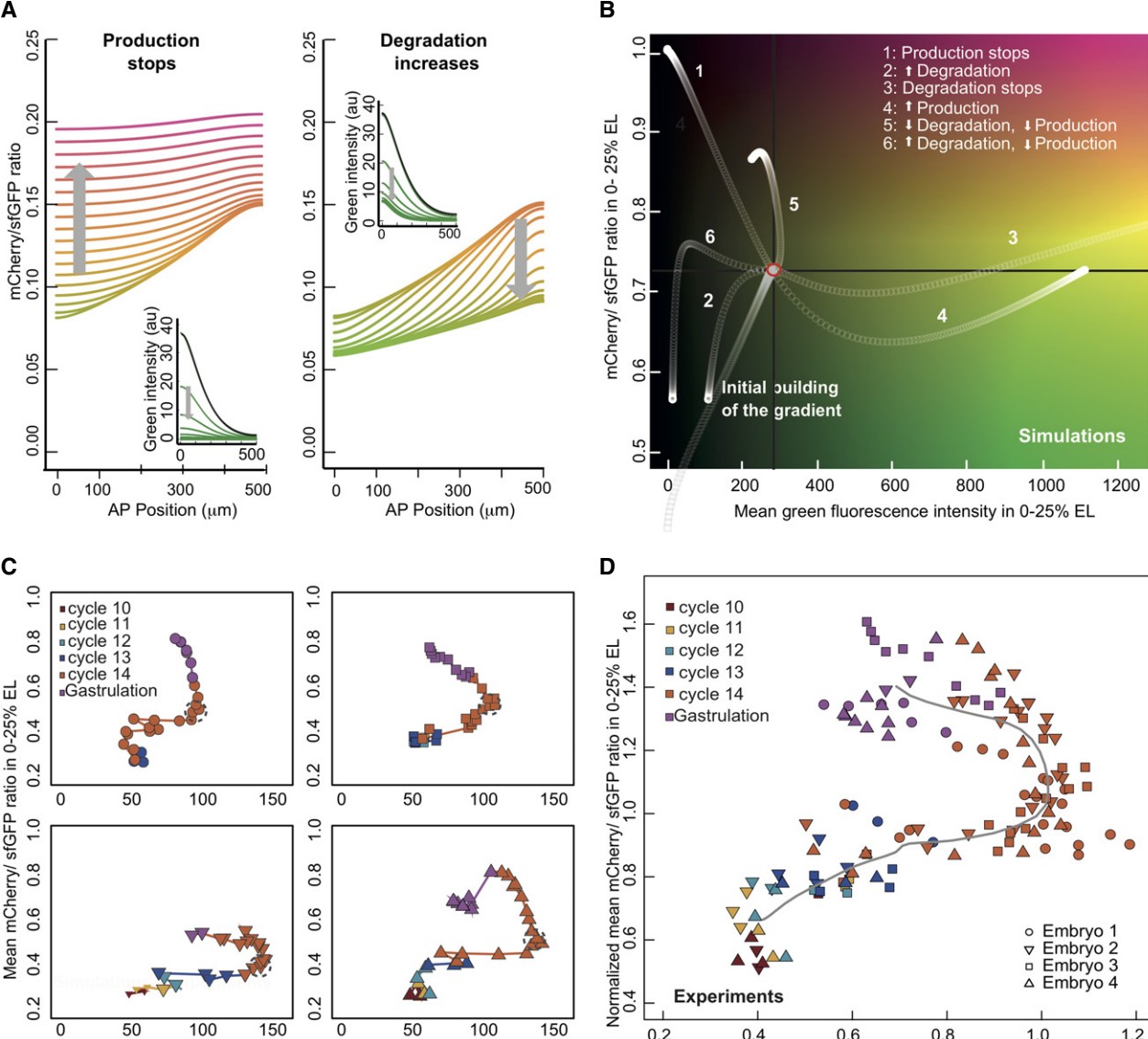

**Figure 6.  At the onset of cellularization Bcd production and degradation rates decrease.**

A    Simulation of SDD model, where after reaching steady state (black line, insets), either production was stopped (left) or degradation increased (right). The gray arrow shows the direction of time, and the color of the lines reflects the mCherry/sfGFP ratio (*y*-axis).

B    As (A), where the SDD model was run until reaching steady state (red circle), and then, production and/or degradation parameters were perturbed (see legend in the plot). The *y*-axis shows the mean mCherry/sfGFP ratio (proxy for average protein age) in the anterior region of the embryo (0–25% of embryo length), and the *x*-axis represents the mean green fluorescence intensity in the same region (age/levels diagram).

C    During the cleavage divisions, the embryos define reproducible trajectories in the age/levels diagram. Experimental data from quantification of movies of y/w embryos as in Fig 4. The trajectories of four embryos from n.c. 10 to gastrulation are shown (each embryo depicted with a different symbol and in a different panel). The time resolution of the time-course was 6 min. The data points are colored according to the division cycle. A time window of approximately 30 min where the trajectories stay (compatible with a steady state) is marked with dotted gray circles. All imaged embryos that managed to start gastrulation were included in the analysis. Embryo 2 is the same from Fig 4.

D    The normalized trajectories from the embryos in panel (C) overlap. The intensity values were normalized for each embryo to the intensity values at the mid n.c. 14 (showed in gray circles in panel C). The gray line depicts the average trajectory.

Source data are available online for this figure.

during this time (Liang *et al*, 2008). Second, the reducing Bcd production in n.c. 14 is also due to the depleting pool of *bcd* mRNA (Little *et al*, 2011). Finally, by n.c. 14 the nuclei are densely packed, which likely decreases the effective diffusion constant (Muller *et al*,

2013). It will be interesting in future work to dissect how the underlying genetic processes are feeding back into the Bcd dynamics. With improving microscopy (Chen *et al*, 2014; Mir *et al*, 2017) and genetic tools for perturbing (Huang *et al*, 2017) and visualizing

(Bothma *et al*, 2018) Bcd activity, the challenge of extracting dynamic information from the early *Drosophila* embryo is increasingly tractable.

Related to the above points, the tandem reporter enables a characterization of the stability of the system. We see that the steady state that was proposed for n.c. 13 and early n.c. 14 based on single-color imaging is only transiently stable, and the Bcd gradient never really achieves an equilibrium. Therefore, we see that the embryo must be interpreting a dynamic gradient at all times. Models that assume a relatively static morphogen profile—such as the canonical French Flag model—are inconsistent with this result. This suggests that more complicated modes of interpretation—likely via spatial and temporal integration through the downstream gap gene network—are essential for reliable interpretation of the Bcd concentration gradient into precise positional information (Gregor *et al*, 2007a; Erdmann *et al*, 2009; Little *et al*, 2013).

It is becoming increasingly apparent that understanding of developmental patterning requires access to both spatial and temporal information (Alexandre *et al*, 2014; Kicheva *et al*, 2014). The tFT reporter enables exploration of morphogen dynamics and gradient formation across multiple temporal and spatial scales. Specific advantages include the following: (i) the analysis is independent of morphogen concentration (though it is sensitive to temporal changes in production); (ii) the tFT reporter decouples degradation and diffusion in model fitting; (iii) image analysis is straightforward (after careful background subtraction) as the tFT ratio is largely independent of imaging inhomogeneity; and (iv) data are extracted from a single time point without need for extended live imaging (and thus avoiding imaging artifacts such as photobleaching). Furthermore, the tFT reporter is sufficiently sensitive to probe degradation regulation. Combined with advances in light-sheet microscopy (Krzic *et al*, 2012; Chen *et al*, 2014; Mir *et al*, 2017), we can now explore the underlying dynamics in forming morphogen gradients *in vivo*.

## Materials and Methods

### Plasmids and fly lines

All plasmids used in this study were cloned by standard methods. For Bcd constructs, all the 3′ and 5′ regulatory regions were conserved. The fluorescent protein fmCherry was evolved in the laboratory and has the following amino acidic changes in comparison with mCherry: K52R, K97N, K126R, K143C, K144R, S152T, Y156H, E158G, N201D, T207L, I215V, D232G. The lines 19.C5 (mCherry-sfGFP-Bcd) and the control 79.fd (mCherry-sfGFP) were done by p-element transformation and were later mapped to the chromosome X and III, respectively, by crossing them with different balancers. All the other lines were generated by landing-site transgenesis to the chromosome II on a fly strain with background y[1] M{vas-int.Dm}ZH-2A w[*];M{3xP3-RFP/attP'}ZH51C. The wt y/w, and the Bcd^E1 and fsd^KG02393 mutant flies were obtained from the Bloomington *Drosophila* Stock Center (stock numbers 42235, 1334, and 12983) and were later crossed with the line 19.C5. The primers used for checking the FSD mutation are as follows: (for) ggcacttgaacagagttacca, (wt specific rev) ggtgaggtaaatttgcactgc and (mutant specific rev) aacaggacctaacgcacagt. Fly stocks

were maintained at room temperature on standard agar–cornmeal–yeast media.

### Western Blotting

Flies of the corresponding lines were caged 2–3 days before embryo collection. Three 45 min pre-laids were performed, followed by 1-h lays. Plates with "Stage 4" embryos (where the Bcd gradient is building up) were left at RT for 70 min, and plates with "Stage 5" embryos (where the Bcd gradient is being degraded) for 140 min. Embryos were then collected and snap-frozen in liquid nitrogen. Protein extraction was carried out in High-Urea buffer (6 M Urea, 5% SDS, 100 mM DTT, 5 mM EDTA, 100 NaPO4 buffer, pH 6.8) at 65°C for 10 min., to a final concentration of 1 embryo/µl. The gels were blotted, and blots were developed with an anti-GFP antibody (rabbit polyclonal, ab6556; Abcam, Cambridge, UK).

### Imaging of live embryos

For embryo collection, 30- to 60-min lays were performed on apple juice-agar plates. The embryos were allowed to develop for 30–60 min at room temperature and dechorionated by a 40-s incubation in 50% bleach. Embryos of the desired age were chosen by visual inspection under a microscope and mounted on a Gelrite gel (Sigma-Aldrich) column inside a glass capillary (Brand 100 ml). Before inserting the capillary in the microscope for imaging, a short segment of the gel containing the embryo was pushed out of the capillary. The chamber of the microscope was filled with PBS. Images were taken using a custom-made light-sheet microscope (Krzic *et al*, 2012), using confocal mode (de Medeiros *et al*, 2015) with a slit size of 30 pixels. In all cases, two stacks were acquired simultaneously from two opposing cameras, and then, two more stacks were taken after a 90° rotation. The illumination of the sample was performed simultaneously from both sides with 10× objectives, and the images were acquired with 25× 1.1NA water-dipping objectives. With this setup, the full embryo fitted inside the field of view and a full stack of the embryo corresponding to 201 *z*-slices (spacing of 1 µm) was acquired. This typically required 30 s (exposure time 100–150 ms). See Appendix Fig S5G–I for demonstration of microscope sensitivity and noise [observed fluctuations in Bcd signal were comparable with those previously reported (Gregor *et al*, 2007a)] and uniformity of illumination across the field of view.

For analysis of single time points, the images were collected 5–10 min after mitotic division 13, when the Bcd gradient is relatively stable (Gregor *et al*, 2007b; Little *et al*, 2011). For time-course analysis, the embryos were collected in 30 min to 1-h-long lays and dechorionated approximately 30 min later. Then, they were taken to the microscope room, at 18°C, and imaging started approximately 15 min afterward. The different experiments were performed with a 4- to 6-min time resolution, to reduce the effect of photobleaching and phototoxicity. All embryos used in the analysis gastrulated normally.

### Image processing and analysis

The four images obtained (corresponding to the four views of each time point, see above) were then fused to a single image (Krzic *et al*, 2012). Autofluorescence and background signals were

removes by linear unmixing (Dickinson *et al*, 2001; Kraus *et al*, 2007) Appendix Fig S13. In brief, the images were corrected by the subtraction of a weighted correction image, taken for this purpose. These images were collected in every experiment as a third channel, using a 488 nm laser and a 594 long-pass filter. The channel was chosen for showing a highly correlated fluorescence intensity to the autofluorescence in the green channel (Appendix Fig S13). The correction images were weighted by a factor calculated for each embryo based on the signal in the end of the posterior pole, where no tFT-BCD fluorescence was detected (Appendix Fig S13).

To quantify the fluorophore intensities and the tFT-Bcd ratio, we performed stereographic projections of the embryo cortical surface (Krzic *et al*, 2012; Schmid *et al*, 2013), Appendix Fig S5A and B. In such a two-dimensional projection, image segmentation was straightforward using Ilastik (Sommer *et al*, 2011) and we extracted the Bcd intensity profile across the whole embryo. We confirmed the projection accuracy by mapping the coordinates back to the three-dimensional data. The embryo images shown (after autofluorescence correction) are made from a *z*-projection of half the embryos using the brightest-point method in Fiji. Displayed images were processed differently, although come from the same original data. Instead of two-dimensional stereographic projections of the embryo cortex, three-dimensional embryo "shells" are shown. The embryo shells are 3D images of the embryos where the interior, corresponding to the yolk, has been erased and only the cortex remains. The reason for doing this is that all the processes we study occur in the cortical region, and the yolk is rich in highly autofluorescent granules that degrade the image clarity. The process for producing embryo shells involves the following steps: (i) The embryo surface is segmented (we do this semi-manually), (ii) the embryo shape is approximated to a cylinder and the central axis is determined, (iii) the coordinate system is changed accordingly to polar coordinates, (iv) the radius of the yolk-cortex limit is found, and (v) all data in smaller radius than this limit are deleted.

Intensity-weighted ratiometric images were used for display purposes only and were generated as outlined previously (Khmelinskii & Knop, 2014). Briefly, the red image was divided by the green one to produce a gray-scale ratiometric image. Then, a LUT of choice was applied and the image was converted to RGB. Each RGB channel was multiplied independently by the green image to bring the intensity information back. Finally, the three channels were converted to tiff to get the intensity-weighted ratiometric image.

### Embryo injections

For the embryo injections, we caged newly hatched flies and collected embryos 2–3 days later. After at least two 45 min pre-lays, we performed 30-min lays, followed by a 30-min incubation at room temperature, and a 40-s dechorionation in 50% bleach. The embryos were mounted for injections on a coverslip with heptane glue. Treated and control embryos were mounted side-by-side. After a 5-min de-hydration, they were covered with a drop of halocarbon oil 700/27 (1:2; Sigma-Aldrich). The injection needles were pulled from borosilicate glass capillaries (1.2 mm outer diameter, 0.94 mm inner diameter; Harvard Apparatus), using a P-97 Flamming/brown puller (Sutter Instrument). The injections were performed using an Eppendorf microinjector model 5242. The injection time was 0.5 s

in every case, and the injection pressure was calibrated for each needle and liquid to produce a drop of the same approximate volume (by simple observation in comparison with the size of a mesh, aiming to a 5% of the embryo's volume). The injections were performed in the posterior pole of the embryos. For the proteasome inhibitor MG132 (C2211; Sigma-Aldrich), we used a concentration of 1 mM in DMSO. Control embryos were injected with DMSO alone. For the RNA, a concentration of 100 ng/μl in RNAses free water was used (controls were injected with water). More details on the determination of the maturation rates of the fluorophores are available in the Appendix.

### Fitting of models

To fit the tFT ratio, we needed to account for the different intensity of the sfGFP and mCherry fluorophores. For this, we considered two options: (i) use *a priori* information and normalize the profiles at large distances (as in Fig 2E inset), where we expected only *old* protein; (ii) or leave the intensity normalization factor as a free parameter. The consequence of the latter scenario is to increase the uncertainty range on the diffusion and degradation parameters. All results presented in the Fig 3 use an additional parameter to account for intensity scaling.

For each model, we calculated $r^2 = r_{sfGFP}^2 + r_{ratio}^2$ where $r_O^2 = \frac{1}{N\langle O \rangle} \sum_{i=1}^{N} (E_i - O_i)^2$ and $E_i$ and $O_i$ represent the model prediction and experimental observation for each position $x_i$, and $< O >$ is the average observed value for each data set. The normalization balances the contribution from the sfGFP profile and the tFT reporter ratio. Minimization of $r^2$ was performed in Matlab using *fminsearch*. Alternative fitting measures, such as the $\chi^2$ measure, did not significantly alter the main conclusions. Fitting the models to only the tFT reporter ratio resulted in errors in fitting to the sfGFP profile and *vice versa*. Hence, all data shown are to model fits to both the sfGFP profile and the sfGFP/mCherry ratio. Fits were also performed using different approximations for FRET effects, Appendix Fig S7I and J.

### Simulations and modeling

Mathematical details of all models considered are described in the Appendix. Briefly, models were numerically solved with Bcd insertion at $x = 0$ μm and no-flux boundary condition at $L = 500$ μm, Appendix Fig S1 for further simulation results. All calculations and simulations for Figs 1–4 were performed in Matlab. The simulations for Fig 6 were performed in R, using the package Reactdiff. Details of analytical approaches to calculating the protein age are described in the Appendix.

**Expanded View** for this article is available online.

### Acknowledgements

We thank the members of the Knop and Hufnagel laboratories for helpful comments and discussions. In particular, we acknowledge technical help from Gustavo de Medeiros, Bálint Balázs, Marvin Albert, and Birgit Besenbeck. We thank the mechanical and electronics workshop of the European Molecular Biology Laboratory (EMBL) for customized hardware. The EMBL Advanced Light Microscopy Facility is acknowledged for support in image acquisition and analysis. This work was supported by the European Molecular Biology

Laboratory (L.D., T.S., T.E.S., and L.H.). T.E.S. was further supported by the EMBL Interdisciplinary Post doc Programme (EIPOD) under Marie Curie Actions COFUND, support from the Kavli Institute for Theoretical Physics, Santa Barbara, and a Singapore National Research Foundation Fellowship. LD was supported from a post doc EcTOP2 fellowship from the CellNetworks cluster of the excellence initiative of the German research council. L.D., M.K., and L.H. acknowledge support by the CellNetworks, University of Heidelberg, Germany, in the context of the EcTop2 project. The authors gratefully acknowledge the data storage service SDS@hd supported by the Ministry of Science, Research and the Arts Baden-Württemberg (MWK) and the German Research Foundation (DFG) through grant INST 35/1314-1 FUGG.

## Author contributions

LD, MK, TES, and LH designed the study, discussed results, and wrote the manuscript. LD, DK, TS, SR, and TES carried out experiments; LD, IK, and TES analyzed the data.

## Conflict of interest

The authors declare that they have no conflict of interest.

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
