## [Review Process File · Molecular Systems Biology]

Bicoid gradient formation mechanism and dynamics revealed by protein lifetime analysis

Lucia Durrieu, Daniel Kirrmaier, Tatjana Schneidt, Ilia Kats, Sarada Raghavan, Lars Hufnagel, Timothy E Saunders and Michael Knop.

Review timeline:

Submission date:	3 rd April 2018
Editorial Decision:	22 nd May 2018
Revision received:	6 th July 2018
Editorial Decision:	25 th July 2018
Revision received:	6 th August 2018
Accepted:	7 th August 2018

Editor: Maria Polychronidou.

Transaction Report:

1st Editorial Decision

22nd May 2018

Thank you again for submitting your work to Molecular Systems Biology. We have now heard back from two of the three referees who agreed to evaluate your study. Unfortunately, after several reminders, we have not managed to obtain a report from reviewer #1. In the interest of time, we decided to proceed with making a decision based on the two available reports. As you will see below, the reviewers think that the application of the timer protein to a morphogen gradient system is an elegant approach and they appreciate the carefully performed quantitative analyses. They raise however a series of concerns, which we would ask you to address in a revision.

The reviewers' recommendations are rather clear therefore I think that there is no need to repeat the points listed below. Please let me know in case you would like to discuss further any of the reviewers' comments.

REFeree REPORTS.

Reviewer #2:

This manuscript by Durrieu et al addresses the fundamental question of the mechanism of Bicoid gradient formation in the early *Drosophila* embryo. The question has received a lot of attention and has led to several proposed models. In this manuscript the authors make use of a new tool - Bicoid fused to an engineered fluorescent timer, which allows them to determine the age of Bicoid protein in space and over time. The authors show that this parameter can be used to distinguish between four possible previously published models of Bcd gradient formation: SDD, nuclear shuttling, RNA gradient and RNA diffusion. Their main conclusion is that the SDD model is the one that most closely recapitulates the data at nuclear cycle 14. In addition, they also find that to explain the

observed temporal dynamics of the Bicoid profile and protein age, the synthesis and degradation rates of Bicoid have to change over time.

Altogether, the novel observations in the manuscript based on the tFT-Bcd and high quality imaging, combined with rigorous quantitative analysis, form an important contribution to the Bicoid field, and will also be of interest to biologists studying morphogen gradient formation in other systems. However, some aspects of the analysis that I outline below are still missing or need better explanation. Overall, I find the manuscript poorly written - many references to figures or precise pointers to Supplementary information are missing, there are multiple imprecise or curtailed explanations. Some key points are noted below, but the writing could be improved beyond these.

Main points:

- The finding that synthesis and degradation rates change over time is a key prediction of the study, however, there is no independent validation of this result. It would significantly strengthen the conclusion if the finding can be corroborated with a FRAP, photoconversion or alternative experiment at different developmental time points. Furthermore, the authors should provide an explanation of the potential causes of the time-dependence, at least at the level of discussion.
- It is not clear whether the consideration of time-dependence in the synthesis and degradation rates could affect the fit to the shuttling, RNA gradient and RNA diffusion models. The authors should comment on and address this point.
- As a key result, Fig. 3a should be extended or amended to demonstrate the sensitivity of the result on the model parameters. The comparison to Fig. 1e in the main text is obscure (for example Fig. 1e and 3a are in different units; Fig. 3a does not contain information about parameter ranges.)
- It is unclear how the chosen maturation rates of mCherry and sfGFP affect the simulated mCherry/sfGFP ratio in the different models. Fig 3a, which captures the main result of the study, shows a minor difference in the ratio between the SDD and shuttling models. However, it is unclear whether the better fit of the experimental data to the SDD model depends on the choice of maturation rates. The authors should further provide a systematic explanation of how they connect the two step mCherry rate to an effective one of 50 min in the main text. And why do they use 20 min for sfGFP in the main text and report 27 min in the Supplementary information?
- The authors should explain how the brightness (quantum yield, extinction coefficient) of the fluorophores are taken into account in the model fitting.
- Fig. S5 does not contain data on the fmCherry maturation rate, as stated in line 171 of the main text. The rate should be measured in a comparable way to mCherry and reported.

Additional points:

- Fig. 1e - The authors should explain what parameter ranges are considered and how the standard deviations are derived. They should also comment on the nearly absent sd of the shuttling model. fig 1d and e - explanation of what is plotted is needed.
- Line 180 - I guess fig. S4m,n is meant. A more detailed explanation of the effects of normalization should be provided within the main text, in particular because Fig 2 ii contains a normalized graph but the rationale behind this is not made clear.
- Fig. 4 - it would be useful to add plots of the intensities and ratio over time for several positions.
- Fig 5a - which orange line?
- It is unclear how and when the production changes in Fig. 5d - the text, figures and figure legends seem contradictory. In the text, it says that the production and degradation decrease (line 295), in the fig. legend to fig. 5d it says that the production increases gradually. Having the relevant plot of the

rates that were used next to Fig. 5d would be useful to clarify this.

- Parameter units should be provided in supplementary tables.

Reviewer #3:

This manuscript presents a quantitative study of the formation of the Bicoid morphogen gradient in the early fly embryo. It's a collaboration between a quantitative imaging lab and the lab that pioneered the tFT reporter's use as a protein age tracer. Consequently, the study is technically impeccable, on par with the state-of-the-art of the Bicoid system. It's the first time this tracer has been used in the fly embryo and a beautiful advertisement for its usage as a quantitative tool. The study confirms many of the known features of the system (1, 2, 3, and 5 in the discussion) and adds the finding of proteasome mediated Bicoid degradation (4). It is a valuable contribution and addition to the field that is interested in the quantitative aspects of Bicoid gradient formation, and it certainly sets a new standard for that field. That being said, however, since its first usage in the early 2000's, the SDD model has in a way always been the golden standard in this field and never been seriously challenged. As such, the novelty and biological relevance of the current findings are somewhat limited. Beyond that, I am uncertain about its broader reach for a general audience.

I have three technical comments:

1. The authors might consider adding to the discussion a paragraph that treats the issue of steady-state in greater depth and what their findings might add to that aspect of the dynamics. The introduction mentions briefly that there is an ongoing debate, and the SDD model can incorporate time dependent parameters, but what have we learned from the current study about the fact that the gradient is never at steady-state and its implications?
2. It was also not entirely clear from reading the manuscript how the authors thought about the degradation properties of Bicoid versus the fusion protein versus the tFT alone. Ultimately, we're interested in Bicoid's properties alone. But how close did we get to that in the current study? Is there still a correction necessary from what was measured with the fusion protein versus the Bicoid protein alone?
3. Shells of embryos is not a nomenclature used in the field. If you need to use it, it should be defined.

Reviewer #2:

However, some aspects of the analysis that I outline below are still missing or need better explanation. Overall, I find the manuscript poorly written - many references to figures or precise pointers to Supplementary information are missing, there are multiple imprecise or curtailed explanations. Some key points are noted below, but the writing could be improved beyond these.

We have improved the writing in the revised version.

Main points:

- The finding that synthesis and degradation rates change over time is a key prediction of the study, however, there is no independent validation of this result. It would significantly strengthen the conclusion if the finding can be corroborated with a FRAP, photoconversion or alternative experiment at different developmental time points.

We agree that further work would be needed to validate whether there are temporal changes in production or degradation during development. Within the framework of this project, it is not feasible to experimentally address these points fully – as this would require significant new work that would represent a paper in itself. Therefore, we have re-focused the results section onto aspects of our results that are clearly supported by our data (pages 20 and 21, lines 460-473), and we have moved the more hypothetical aspects to discussion (page 23, lines 503-514). In addition, we have performed a proteasome inhibitor experiment to investigate the effects of stopped degradation at different stages of development. While we see a clear impact of the inhibitor at stage 4, we do not see a strong effect when injected during cellularization (see Figure below). This result is consistent with the idea of halted/slowed degradation at this stage. However, we are uncertain about the extent to which cellularization has been completed in these embryos and whether this hinders access of the inhibitor to the cells. Given the situation that the literature is very divergent on the degradation rates and their changes during cycle 14**, we decided to remove any conclusions about degradation rates at this very specific time point, and to restrict our interpretation to the clearer results.

**–Liu *et al.*¹ determined the strength of the degradation by incubating purified Bcd protein with extracts from 0-1, 1-2 or 2-3 hrs old embryos that had been treated with CHX to inhibit translation. Bcd levels were then assessed in Western Blots. They find that degradation is similar in 0-1 and 1-2 hr embryo extracts, and then slows down significantly in 2-3 hr old embryos (this period corresponds to the cellularization of the blastoderm, during cycle 14). Drocco *et al.*², measured the Bcd degradation rate by a method based on photoconversion of Dronpa-Bcd on live embryos, and found it increases drastically at the beginning of cycle 14. Therefore, both of these approaches predict time-varying degradation rates, but with opposing reports for the behaviour of the Bcd lifetime in cycle 14.

Reviewer Response Figure 1. Bcd proteasomal degradation decreases in Stage 5 embryos. Stage 4 or Stage 5 embryos expressing the tFT-Bcd construct were injected with MG132 (orange) or DMSO (blue), and imaged on a confocal microscope 30 min later. Average GFP levels in the anterior third of the embryo were quantified from single, equatorial z-slices. The symbols indicate individual embryo measurements, and the lines signal the median of the distribution. The gray brackets illustrate the difference in GFP fluorescence at each embryonic stage. Statistical significance was assessed by a two-sample Wilcoxon test (Stage 4 p-value = 0.01, Stage 5 p-value = 0.30).

Furthermore, the authors should provide an explanation of the potential causes of the time-dependence, at least at the level of discussion.

We now introduce explanations for the temporal trajectory of the Bcd age measurements in the Results (page 19, lines 440-442), and Discussion (page 23, lines 528-540).

- It is not clear whether the consideration of time-dependence in the synthesis and degradation rates could affect the fit to the shuttling, RNA gradient and RNA diffusion models. The authors should comment on and address this point.

The fittings for model selection in Figure 1 were performed on data from embryos in

early nuclear cycle 14, assuming that the gradient was in steady state. This assumption seems to be challenged by our later results in Figure 6, where we observe that the production and degradation rates change during this stage of development. However, we do find a short period during nuclear cycle 14 where the system is roughly in steady-state – identified by the average Bcd concentration and age remaining relatively constant- and the fittings were done in this regime. We have added a panel, Figure 6C, where we highlight this observation, and a paragraph discussing it on the main text (pages 20-21, lines 466-473). To illustrate this, we have plotted the data from Figure 6C and D in a new format, that shows that the ratio is relatively constant for a period of time between nuclear cycle 13 and 14, presumably because the system is in a quasi-equilibrium. Of course, this steady-state is much shorter than previously thought, which could have consequences in the gradient interpretation, and as well on our fittings. See lines 542-551 (pages 23 and 24) for discussion in the manuscript.

We agree that introducing time dependence in these models would alter the fit quality. However, for the RNA gradient and diffusion models, it would still not be possible to fit the protein age using reasonable parameters – as these models predict curve shapes that are fundamentally different from what we measured. In the shuttling model degradation is not present, so a time-varying degradation rate cannot be included. The fact that we find that there is significant degradation and that this impacts the gradient is evidence against the idea that nuclear shuttling is sufficient alone for gradient formation. Of course, there is also time-varying Bcd production. However, for relatively slow changes in production this is unlikely to have a significant effect on protein age, as changes in protein production generally lead to only transient shifts in the tFT reporter, as shown in Theory Figure 1 in the Appendix, and the more radical stop of the productions happens in mid-late cycle 14, while the fittings of the models were performed in early cycle 14.

For the shuttling model, it is plausible that the diffusion constant changes over time, due to, for example, a change in the density of nuclei due to their doubling with each cycle. We have re-run the simulations considering decreasing time-dependent diffusion. We consider a simple phenomenological form for this time dependence to minimize additional parameters. We find that such time dependence does increase the fit quality, but the model is still not as good at explaining our observed data as the SDD model. We now include this analysis in Appendix Figure S12 and lines 516-526 (pages 22-23).

-As a key result, Fig. 3a should be extended or amended to demonstrate the sensitivity of the result on the model parameters. The comparison to Fig. 1e in the main text is obscure (for example Fig. 1e and 3a are in different units; Fig. 3a does not contain information about parameter ranges.)

We agree that Figure 3A is a critical result of the paper and requires further details. We have now split Figure 3 into two, with the fitting to the key data in Figure 3 and the perturbations shown in Figure 4. The revised Figure 3 now includes a panel showing the conversion of protein age into the tandem ratio (panel **B**) and a comparison of the fit quality of the different models (panel **C**). We have also extended the model fitting to test the tandem ratio from the fmCherry-sfGFP Bcd-tFT reporter (panel **A** inset). In this case, we used the fitted parameters to the mCherry-

sfGFP-Bcd-tFT reporter and only allowed the red fluorophore maturation rate and relative fluorescence levels to alter. The resulting fit of the SDD model is very good, though we note that the nuclear shuttling model is also a good fit within the experimental error. We have extended the discussion of these results (page 14, lines 305-310) and given further details of the simulation details in the Methods (pages 29-30, lines 693-712). See also response to below comment.

- It is unclear how the chosen maturation rates of mCherry and sfGFP affect the simulated mCherry/sfGFP ratio in the different models. Figure 3a, which captures the main result of the study, shows a minor difference in the ratio between the SDD and shuttling models. However, it is unclear whether the better fit of the experimental data to the SDD model depends on the choice of maturation rates. The authors should further provide a systematic explanation of how they connect the two step mCherry rate to an effective one of 50 min in the main text. And why do they use 20 min for sfGFP in the main text and report 27 min in the Supplementary information?

The maturation rates are important parameters in the data fitting. We now more rigorously incorporate uncertainty in the maturation rates into our model fitting. In Figure 3B, we simulate the effect of varying the maturation rates on the subsequent tandem reporter ratio. For the range 20-50 minutes – which is close to the apparent Bcd lifetime – the variability is largest. As we now show more clearly, this does not alter our key conclusion that the SDD model is the best fit to the data, since the predictions of the alternative models are qualitatively distinct. Importantly, the maturation times of sfGFP, mCherry, and fmCherry are all quite distinct so the tandem reporter works well. However, the uncertainty on the estimated maturation rates does increase the variability on our predictions for the Bcd (effective) diffusion coefficient and lifetime. In Figure 3C, we now show the fit quality for a range of fluorophore maturation rates.

We estimated a maturation time for sfGFP of 27 ± 2 min. The error in the estimated rates was assessed by likelihood profile analysis⁴ and is now shown in Appendix Figure S6D-F. So long as for the three fluorophores used the maturation times are $\tau_{fmCherry} < \tau_{sfGFP} < \tau_{mCherry}$, our key result that the SDD model is the best fit to the data likely remains correct.

In all the fittings of the tFT-Bcd gradient, we used a two-step model for mCherry maturation ($T_1=40$ min, $T_2 = 9$ min). We have clarified the text in lines 281-284, as it was incorrect to infer this implies a maturation time of ~50 min, but we note that this did not affect our simulations as the chemical kinetics were correctly implemented in the fitting algorithm. The global maturation rate of mCherry is expected to be close to the limiting step of 40 min. We thank the reviewer for pointing out this mistake in the text.

- The authors should explain how the brightness (quantum yield, extinction coefficient) of the fluorophores are taken into account in the model fitting.

The brightness of the fluorophores is different between sfGFP and mCherry. However, precisely measuring this is challenging. When fitting the data, we have a scaling parameter, which effectively accounts for differences between the brightness

in the fluorophores. In our original submission we used the intensities of the sfGFP and mCherry profiles in the posterior (where we expect them to be similar) to estimate this parameter. However, this approach introduced bias, and so now the scaling parameter is kept as a fully free fitting parameter within the model. As with the uncertainty in the fluorophore folding rates, including such a fitting parameter does not alter our key conclusions, but it does increase the uncertainty in our estimations of the dynamic parameters. We have clarified this issue in the manuscript Results section (page 14, lines 305-310 and 319-320) and Methods (pages 29-30, lines 693-712)

- Fig. S5 does not contain data on the fmCherry maturation rate, as stated in line 171 of the main text. The rate should be measured in a comparable way to mCherry and reported.

We have now included the fmCherry maturation rates estimation in the Appendix Figure S6 (C and F-G). We have slightly expanded the description of fmCherry in the text (page 9 lines 185-187, page 12 255-257 and 272-275, page 13 291-294), and we have separated and expanded the figure on the maturation rates determination (Appendix Figure S6), showing now the likelihood profile analysis to assess the quality of the parameters estimation. Additionally, we have also included fitting of the alternative models for Bcd gradient formation to the tFT fmCherry-sfGFP-Bcd reporter data (Figure 3A inset).

Additional points:

- Fig. 1e - The authors should explain what parameter ranges are considered and how the standard deviations are derived. They should also comment on the nearly absent sd of the shuttling model. fig1d and e - explanation of what is plotted is needed.

We have extended the description of this figure, including discussion of errors. Lines 132-161, on pages 7-8.

- Line 180 - I guess fig. S4m,n is meant. A more detailed explanation of the effects of normalization should be provided within the main text, in particular because Figure 2 ii contains a normalized graph but the rationale behind this is not made clear.

As detailed above we have significantly extended our parameter fitting discussion (page 14 lines 305-310 and 319-320, pages 29-30 lines 693-712). We have also streamlined the Appendix Figures to make sure they are clearer in substantiating the results presented in the paper.

- Fig. 4 - it would be useful to add plots of the intensities and ratio over time for several positions.

We have included the data as suggested (Figure 5C) and updated the text (pages 17-18, lines 419-424).

- Fig 5a - which orange line?

Fixed (page 41, line 1002).

-It is unclear how and when the production changes in Fig. 5d - the text, figures and figure legends seem contradictory. In the text, it says that the production and degradation decrease (line 295), in the fig. legend to fig. 5d it says that the production increases gradually. Having the relevant plot of the rates that were used next to Fig. 5d would be useful to clarify this.

Thank you for the suggestion. Now the panels relating to the parameter changes are together with the former Fig 5d in the Appendix Figure S11 (see our response to the first point). We have also corrected the mistake in the figure legend.

- Parameter units should be provided in supplementary tables.

Done as requested.

Reviewer #3:

I have three technical comments:

1. The authors might consider adding to the discussion a paragraph that treats the issue of steady-state in greater depth and what their findings might add to that aspect of the dynamics. The introduction mentions briefly that there is an ongoing debate, and the SDD model can incorporate time dependent parameters, but what have we learned from the current study about the fact that the gradient is never at steady-state and its implications?

We thank the Reviewer for this suggestion. We have now included a new figure panel that focus more on the issue of the steady-state (Figure 6C), and an explanation in the text (pages 20-21, lines: 466-473). Further, we have included a more detailed analysis in the Discussion, and linked our results with the broader debate about the dynamic state of biological systems, lines 542-551 (pages 23-24).

2. It was also not entirely clear from reading the manuscript how the authors thought about the degradation properties of Bicoid versus the fusion protein versus the tFT alone. Ultimately, we're interested in Bicoid's properties alone. But how close did we get to that in the current study? Is there still a correction necessary from what was measured with the fusion protein versus the Bicoid protein alone?

This is an important point and we agree with the reviewer that in the original submission these points were not sufficiently clear. Indeed, all our conclusions relate to tFT-Bcd. However, since the fusion protein is fully functional and embryos develop normally, it is reasonable to assume that we are studying an intact system and that differences between the tagged and the untagged Bcd are likely secondary effects.

Specifically:

1) The construct rescues the null phenotype, so it is fully functional, which for Bcd also means a "normal" gradient (Appendix Figure S3).

2) The measured gradients obtained from our constructs are comparable to previously published quantifications of the Bcd morphogen gradient.

3) The timing of maximum Bcd-sfGFP intensity is similar to previous reports³. Therefore, we are confident that the dynamics of our construct are close to wildtype conditions.

4) The control line with the tandem fluorescent timer without the Bcd protein has a completely different behavior, with very low degradation (Figure 2E).

We have now extended our discussion of the controls regarding the functionality of our tFT-Bcd reporter, lines 189-190, 198-204, 215-217, and a summary in lines 219-225 (pages 9-10).

3. Shells of embryos is not a nomenclature used in the field. If you need to use it, it should be defined.

Yes, this is a new nomenclature. We have now added a clearer definition in the figure legend (page 38, lines 933-936) and a more complete description on how they are produced on the Methods (page 28, lines 651-662). The “shells” are a way of processing 3D images for display purposes. They consist on 3D reconstructions of embryos where the external layer - the cortex - is kept, but the interior - the yolk - is digitally removed. We do this because all the events we study happen in the cortical region, and inclusion of the yolk adds noise and increases the image size.

Reviewer Response References

1. Liu, J., He, F. & Ma, J. Morphogen gradient formation and action. *Fly* **5**, 242–246 (2011).
2. Drocco, J. A., Grimm, O., Tank, D. W. & Wieschaus, E. Measurement and perturbation of morphogen lifetime: effects on gradient shape. *Biophys. J.* **101**, 1807–1815 (2011).
3. Little, S. C., Tkačik, G., Kneeland, T. B., Wieschaus, E. F. & Gregor, T. The Formation of the Bicoid Morphogen Gradient Requires Protein Movement from Anteriorly Localized mRNA. *PLoS Biol* **9**, e1000596 (2011).
4. Raue, A. *et al.* Data2Dynamics: a modeling environment tailored to parameter estimation in dynamical systems. *Bioinformatics* **31**, 3558–3560 (2015).
5. Kavousanakis, M. E., Kanodia, J. S., Kim, Y., Kevrekidis, I. G. & Shvartsman, S. Y. A compartmental model for the bicoid gradient. *Dev. Biol.* **345**, 12–17 (2010).
6. Spirov, A. *et al.* Formation of the bicoid morphogen gradient: an mRNA gradient dictates the protein gradient. *Development* **136**, 605–614 (2009).
7. Dilão, R. & Muraro, D. mRNA diffusion explains protein gradients in Drosophila early development. *J. Theor. Biol.* **264**, 847–853 (2010).

Thank you for sending us your revised manuscript. We have now heard back from reviewer #2 who was asked to evaluate your study. As you will see below, this reviewer is satisfied with the modifications made and thinks that the study is now suitable for publication.

Before we formally accept the study for publication, we would ask you to address the following remaining editorial issues:

REFEREE REPORTS.

Reviewer #2:

The authors have improved the manuscript and addressed my concerns.

Corresponding Author Name: Prof. Michael Knop

Journal Submitted to: The EMBO Journal

Manuscript Number: MSB-18-8355